# Effects of Dietary Protein Levels on Production Performance, Meat Quality Traits, and Gut Microbiome of Fatting Dezhou Donkeys

**DOI:** 10.3390/microorganisms13061388

**Published:** 2025-06-14

**Authors:** Yunpeng Wang, Keqiang Diao, Han Li, Chongyu Zhang, Guiguo Zhang, Cuihua Guo

**Affiliations:** Key Laboratory of Efficient Utilization of Non-Grain Feed Resources (Co-Construction by Ministry and Province), Ministry of Agriculture and Rural Affairs, College of Animal Science and Technology, Shandong Agricultural University, Tai’an 271018, China

**Keywords:** dietary protein levels, Dezhou donkeys, production performance, meat quality traits, microbiota

## Abstract

This study aimed to investigate the effects of varying dietary protein levels on growth performance, meat quality traits, amino acid and fatty acid compositions, and hindgut microbiota in Dezhou donkeys. Eighteen 12-month-old male donkeys, weighing 188 ± 9 kg, were randomly allocated into three groups and fed diets containing 11.03% (LP), 12.52% (MP), and 14.06% (HP) protein. The average daily gain (ADG) was significantly higher (*p* < 0.05) in the HP and MP groups, while the feed conversion ratio (FCR) was lower (*p* < 0.05) compared to the LP group. The MP group exhibited superior performance in terms of serum albumin (ALB) and high-density lipoprotein (HDL) levels, as well as protein digestibility (*p* < 0.05). Improvements in meat tenderness, as well as increased levels of leucine, flavor amino acids (FAAs), and non-essential amino acids (NEAAs) (*p* < 0.05), were observed in the MP group compared to those in the LP and HP groups. The levels of total fatty acids (TFAs), saturated fatty acids (SFAs), unsaturated fatty acids (UFAs), and monounsaturated fatty acids (MUFAs) were higher (*p* < 0.05) in the LP and MP groups than in the HP group, with no significant differences (*p* > 0.05) observed between the LP and MP groups. The genera *Prevotella*, *Clostridium*_*sensu*_*stricto*_*1*, *NK4A214*_*group*, *Oscillospiraceae*_*UCG*-*002*, and *Oscillospiraceae*_*UCG*-*005* in the rectum were identified as differential microbes associated with varying dietary protein levels. In conclusion, this study indicates that a dietary protein level of 12.52% could enhance the growth performance, dietary nutrient digestibility, slaughter performance, and meat quality of Dezhou donkeys by modulating hindgut microbial communities.

## 1. Introduction

The Dezhou donkey, an indigenous breed strain of herbivorous draft livestock in China, has played an indispensable role in the history of human transportation [1]. However, with the progress of modern transportation, the value of donkeys has gradually shifted towards their utilization in meat production [2] and the healthcare industry [3]. The documentation of global nutrition research on donkeys is insufficient compared to that of other domesticated animals, resulting in the absence of a standardized approach for donkey breeding. This deficiency has negatively impacted the expansion of donkey farming. According to the 2023 China Statistical Report, the donkey population in China has experienced a substantial decline of 35.21% over three years, decreasing from 2.678 million in 2019.

In the fattening process of donkeys, optimizing growth performance and meat quality is the key to improving economic benefits. The effect of dietary nutrition on the growth and production performance of equine animals is indisputable, and the related research primarily focuses on foals [4] and lactating donkeys [5]. Nevertheless, we previously recommended a suitable energy-concentrated feed in order to achieve satisfactory growth and fattening performance of Dezhou donkeys [6]. It is well established that animal production performance is negatively impacted by diets with insufficient protein levels [7]. Conversely, excessively high protein levels can result in resource wastage [8] and environmental pollution [9]. Adequate levels of dietary protein are crucial for meeting the production requirements during the fattening phase of animals and improving the flavor profile of pork and mutton [10,11]. Similarly, an appropriate dietary protein level is also important for Dezhou donkeys. The investigation of appropriate dietary protein strategies to optimize production efficiency in donkeys is critically important, especially given the current protein feed shortage [12].

As demonstrated by previous studies, the composition of the microbiota in the digestive tracts of herbivores can be regulated by changes in dietary protein concentration [13]. This modulation acts as a crucial mediator through which dietary factors impact host health [14], exhibiting a strong correlation with nitrogen utilization efficiency [15] and consequently enhancing performance [16] and improving meat quality [17]. However, the impact of dietary protein levels on the hindgut microbiota in donkeys remains uncertain.

Additionally, the utilization of the pellet diet in herbivores enhanced production performance [18] and reshaped the composition of the gut microbiome [19]. Therefore, the current study employed diets with different protein levels, formulated as total mixed rations (TMRs) and subsequently pelleted, to investigate the protein nutritional requirements of donkeys during the fattening stage. This was achieved by evaluating growth and production performance, meat quality, and amino acid and fatty acid profiles. The findings provide practical guidelines for the scientifically sustainable development of donkey breeding.

## 2. Materials and Methods

### 2.1. Experimental Design and Feeding Management

This study was carried out from August to October 2022 at the Animal Experiment Station of Shandong Agricultural University (36°19′12″ N, 116°14′24″ E), Liaocheng City, Shandong Province, China. All protocols and experimental procedures were approved by the Animal Nutrition Department of Shandong Agricultural University (Protocol number: SDAUA-2022-242). During the experimental period, the temperature ranged from 18 °C to 28 °C, and the relative humidity ranged from 62% to 75%. For the lighting program, the experiment utilized natural daylight, with a photoperiod ranging from approximately 11 h to 14 h each day, and no artificial lighting was employed in this study.

A total of 18 healthy male Dezhou donkeys (12 months old, with an initial weight of 188 ± 9 kg) were randomly allocated to three dietary groups, with six replicates per group, and each animal was fed in an individual pen. All the donkeys in this study were housed individually in stalls. Each stall comprises an indoor area (with one window on the front wall) and an outdoor exercise area, with the two areas connected by a sufficiently large door. The indoor section has a cement floor measuring 4 m (length) × 6 m (width) and is equipped with both a water trough and a feeding trough. The outdoor exercise area measures 10 m (length) × 6 m (width) and includes a sand field (5 m × 6 m) as well as a red-brick paved surface of the same dimensions.

All animals were provided with pelleted TMR diets containing the same digestible energy level (11.8 MJ/kg) but with different crude protein levels of 11.03%, 12.52%, and 14.06% protein separately in three different groups, which were labeled as the LP, MP, and HP groups, respectively. These protein levels were set based on previous studies and the nutrient requirements for adult horses weighing less than 200 kg, as outlined in the NRC (2007) guidelines [20]. The specific formula and nutritional composition followed the feeding standard for Dezhou donkeys in China (DB 37/T 3605-2019) [21], as detailed in Table 1.

The animal experiment lasted 70 d, including a 10 d adaptation period followed by a 60 d formal feeding phase for data collection. The experimental donkeys were fed three times daily at 6:00 a.m., 12:00 p.m., and 6:00 p.m., with ad libitum access to feed and water.

### 2.2. Dietary Nutrient Determination

The feed nutrients, including dry matter (DM) (method 930.15), ash (method 942.05), ether extract (EE) (method 920.39), and crude protein (CP) (method 984.13), were analyzed according to the methodologies described by the Association of Official Analytical Chemists [22]. The fiber-related indices in the sample, such as crude fiber (CF), neutral detergent fiber (NDF), acid detergent fiber (ADF), and acid detergent lignin (ADL) were determined using the methodology established by Van Soest [23]. Other nutrients were calculated using the formulas: Hemicellulose (HCEL) (%) = NDF (%) − ADF (%); Organic matter (OM) (%) = DM (%) − ash (%); Carbohydrate (CH_2_O) (%) = DM (%) − Ash (%) − CP (%) − EE (%); and Nitrogen-free extract (NFE) (%) = DM (%) − Ash (%) − CP (%) − EE (%) − CF (%).

### 2.3. Dietary Nutrient Digestibility Determination

Prior to the conclusion of the experiment, fresh fecal samples were collected from each donkey over three consecutive days. These samples were then thoroughly mixed and air-dried to create a composite sample for each individual donkey. The fecal samples were dried in an oven at 65 °C until a constant weight was achieved. Thereafter, the samples were sieved through a 40-mesh sieve and analyzed using the same methodologies employed for dietary nutrients in this study.

The digestibility was evaluated using the endogenous indicator method, and the endogenous marker selected was ADL. The specific calculation method is as follows: Dietary nutrient digestibility (%) = 100 × [1 − (dietary ADL content/fecal ADL content) × (nutrient content in fecal sample/dietary nutrient content)].

### 2.4. Growth Performance Analysis

The weights of the donkeys were measured before morning feeding at the beginning and end of the formal feeding phase (60 d), which noted the initial body weight (initial BW) and final body weight (final BW), respectively, and the average daily gain (ADG) was calculated according to the formula: (final BW − initial BW)/60. The feed intake and the amount left by each donkey were recorded every day, which were used to calculate the average daily dry matter intake (DMI). Then, the feed conversion ratio (FCR) was calculated by dividing the DMI by the ADG.

### 2.5. Serum Biochemical Analysis

Fasting blood samples were collected from the jugular vein on the day prior to the completion of the animal experiment. After centrifugation at 3000× *g* for 15 min at 4 °C, the resulting serum was promptly stored at −20 °C in a refrigerator. Subsequently, biochemical indices were measured using an automatic analyzer (Hitachi 7020, Tokyo, Japan).

### 2.6. Slaughter Performance

The donkeys were slaughtered using electrical stunning, following the protocols recommended by the Animal Ethics Committee at Shandong Agricultural University. The carcass weight was calculated after removing the total weight of the head, hooves, fur, blood, and viscera from the live weight before slaughter. The dressing percentage was determined by dividing the carcass weight by the live weight.

Two samples (one about 500 g, the other about 4 g) of the longissimus dorsi muscle, located between the 13th and 18th thoracic vertebrae on the left side of the carcass, were promptly collected from each donkey at 24 h postmortem (0 °C–4 °C). The 4 g samples were rapidly put in liquid nitrogen and subsequently stored at a −80 °C freezer for the determination of free fatty acids and amino acids. The 500 g samples were vacuum-packaged and transported to the laboratory at 0 °C–4 °C within 2 h for meat quality analysis. The sarcolemma and attached fat were removed for all the samples.

### 2.7. Meat Quality Determination

#### 2.7.1. pH and Color

The pH of the longissimus dorsi muscle was measured using a portable pH meter (Seven2Go-S2, Mettler Toledo, Switzerland) by inserting the probe 2 cm into the muscle tissue. Prior to measurement, the instrument was calibrated using two standard buffer solutions (at pH 4.00 and 7.00).

After a 45 min blooming period, the meat color parameters, including lightness (L*), redness (a*), and yellowness (b*), were evaluated at three distinct locations on the exposed surface using a Minolta CR-10 colorimeter (Konica Minolta, Tokyo, Japan). Before determining the meat sample color, this instrument was calibrated using a black plate (Model CR-A47) (Konica Minolta, Tokyo, Japan) and a white plate (Model CR-A44) (Konica Minolta, Tokyo, Japan). The measurements were conducted under standard illuminant D65, with a 10° observer angle and an 8 mm aperture.

#### 2.7.2. Meat Water-Holding Capacity and Shear Force Assessment

The longissimus dorsi muscle samples were cut into pieces measuring 10 cm × 5 cm × 1 cm. For the determination of drip loss, the initial weight of each piece of the meat sample was measured and recorded as W1. The samples were then placed in a drip loss tube, ensuring they did not contact the walls, and stored at 4 °C for 24 h. After this period, the surface moisture of the samples was removed using filter paper, and the weight was measured again and recorded as W2. The formula [(W1 − W2)/W1] × 100 was used to calculate the drip loss (%) [24]. To analyze the cooking loss, each longissimus dorsi muscle piece was weighed (W1) and then sealed in a plastic bag and cooked in a water bath at 80 °C until the internal temperature reached 70 °C. Subsequently, these samples were cooled to room temperature. Surface moisture was removed using absorbent paper before reweighing (W2). Cooking loss (%) was calculated as [(W1 − W2)/W1] × 100%. The samples intended for cooking loss determination were stored at 4 °C for 12 h. Thereafter, the cooked samples were cut into 3 cm × 1 cm × 1 cm meat strips parallel to the direction of the muscle fibers and subsequently assessed using an electronic muscle tenderness meter (model C-LM3B, Tenova, Harbin, China).

#### 2.7.3. Chemical Analysis of the Meat

The nutrient composition of the longissimus dorsi muscle, including ether extract, crude protein, and ash, was assessed according to the methods outlined in the National Standards of the People’s Republic of China [25,26,27]. Specifically, the ether extract of the freeze-dried meat was determined using the Soxhlet extraction method, the crude protein content was analyzed through the Kjeldahl determination method, and the ash content was measured by combusting the sample at temperatures ranging from 550 °C to 600 °C in a Muffle furnace.

### 2.8. Determination of Meat’s Free Fatty Acids

In this study, medium–long-chain fatty acids of the meat samples were analyzed by GC-MS (Agilent Technologies Inc., Santa Clara, CA, USA). Free fatty acids were extracted from meat samples through a two-step extraction and derivatization procedure. Approximately 1 g of finely ground longissimus dorsi muscle, obtained through multipoint random sampling from a 4 g sample, was mixed with a chloroform–methanol solution (1:1, *v*/*v*) and a 0.88% NaCl solution. The mixture was centrifuged at 8000 rpm for 15 min at room temperature, and the chloroform layer was transferred to a new tube. The residual NaCl phase was then mixed with 2 mL of dichloromethane and centrifuged again under the same conditions. The dichloromethane layer was collected, combined with the previous chloroform layer, dried under nitrogen, and mixed with a methylation reagent (sulfuric acid/methanol = 1:25, *v*/*v*). This mixture was incubated at 80 °C for 2 h. After cooling, 2 mL of n-hexane and 1 mL of distilled water were added, and the solution was centrifuged at 5000 rpm for 5 min. The supernatant was treated with 1 mL of water and centrifuged again at 5000 rpm for 5 min. The final supernatant was dried under nitrogen and mixed with isooctane for analysis. The GC was performed using an Agilent 6890 with an INU-Sil 88 column (100 m × 0.25 mm × 0.25 µm), 1 µL injection, a split ratio of 10:1, and nitrogen flow of 1.0 mL/min. The column temperature started at 100 °C for 5 min, was increased to 240 °C at 4 °C/min, and held for 15 min. MS detection was conducted by an Agilent 5977 (Agilent Technologies, USA) with an EI source and MassHunter workstation, and the operational conditions were as follows: the injector at 260 °C, the quadrupole at 150 °C, full scan mode, and a quality control scanning range of 30–550 *m*/*z*.

### 2.9. Measurement of Meat’s Free Amino Acids

Each meat sample was extracted with 0.5 mL of 0.5 M HCl solution on a shaker at room temperature and then filtered through a 0.22 µm membrane. Next, 10 μL of each sample was combined with 70 μL of borate buffer and 20 μL of AccQ·Tag reagent (Waters Corporation, item 186003836, Milford, MA, USA) in phosphorus vials. The mixture was heated and cooled before use. Amino acids were detected using ultra-high-performance liquid chromatography (UHPLC, Vanquish, Thermo, Waltham, MA, USA) and high-resolution mass spectrometry (Q Exactive, Thermo, USA). The UPLC system used a Waters BEH C18 column (50 × 2.1 mm, 1.7 μm) at 55 °C, with a 0.5 mL/min flow rate and 1 μL injection volume. The mobile phase was 0.1% formic acid in water (A) and 0.1% formic acid in acetonitrile (B). Mass spectrometric analysis was performed employing electrospray ionization (ESI), the parameters were sheath gas at 40, auxiliary gas at 10, ion spray voltage at +3000 V, vaporizer at 350 °C, capillary at 320 °C, full scan, and positive ionization.

### 2.10. Determination of Gut Microbiota

Rectal content samples for each donkey were taken and immediately preserved in 5 mL sterile tubes. They were then rapidly frozen using liquid nitrogen for subsequent analysis. Gut microbiota analysis was conducted through 16S rRNA gene sequencing, following the methodology described in our previous study [6]. This study comprised four sequential steps, including extraction of genomic DNA, generation of amplicons, quantification and qualification of PCR products, and preparation of libraries for sequencing. Genomic DNA from the gut microbiota was extracted from rectal content samples using the cetyltrimethylammonium bromide (CTAB)/sodium dodecyl sulfate (SDS) method. The V3-V4 hypervariable region of the 16S rRNA genes was amplified using the specific primers (forward primer: 5′-TAGATACCCSGTAGTCC-3′ and reverse primer: 5′-CTGACGRCRGCCATGC-3′). A QIAquick Gel Extraction Kit (Qiagen, Hilden, Germany) was employed to purify the PCR products. The sequencing library was prepared using the TruSeq^®^ DNA PCR-Free Sample Preparation Kit (Illumina, San Diego, CA, USA), and its quality was evaluated using both the Qubit 2.0 Fluorometer (Thermo Scientific) and the Agilent Bioanalyzer 2100 system. Subsequently, the library was sequenced on the Illumina NovaSeq platform, generating 250 bp paired-end reads. The raw paired-end reads were assembled into longer sequences and subjected to quantitative filtering using PANDAseq (version 2.9) to eliminate low-quality reads. The high-quality sequences were clustered into operational taxonomic units (OTUs) at a 97% similarity threshold using UPARSE (version 7.0) within QIIME (version 1.8). Chimeric sequences were identified and removed using the UCHIME (version 4.2). Taxonomic assignments for the OTUs were performed using the RDP classifier against the SILVA 16S rRNA gene database (Release 128).

The raw 16S rRNA gene sequencing data have been deposited in the NCBI database under accession number PRJNA977424. Beta diversity (β-diversity) was analyzed by principal coordinate analysis (PCoA) based on Bray–Curtis distances. Discrepant microbes were identified through GraphPad Prism (version 8.0.1) and linear discriminant analysis effect size (LEfSe) analysis performed via the BIC online tool (https://www.bic.ac.cn/BIC/#/, accessed on 21 April 2025).

### 2.11. Statistical Analysis

In this study, one-way ANOVA was conducted using SPSS 27.0 software (IBM Corp., Armonk, NY, USA), and multiple comparisons were performed using Duncan’s method. The results are presented as mean and standard errors of the mean (SEM), and differences were deemed statistically significant at *p* < 0.05. Apart from the data utilized for LEfSe analysis (which is categorized under multivariate statistical analysis), all other data displayed in the tables or figures were subjected to normality testing using the Shapiro–Wilk test via SPSS 27.0 software (a method appropriate for smaller sample sizes). The *p*-values for the data within each group were all greater than 0.05, thereby confirming a normal distribution.

## 3. Results

### 3.1. Growth Performance of Donkeys

Table 2 displays the impacts of varying dietary crude protein levels on the growth performance of donkeys. There were no significant differences (*p* > 0.05) in the initial BW, final BW, and DMI among the three groups. However, the HP group exhibited significant enhancement (*p* < 0.05) in ADG and FCR compared to the LP group, while no significant difference (*p* > 0.05) was noted when compared to the MP group.

### 3.2. Serum Biochemistry Analysis of Donkeys

Table 3 shows the impact of varying dietary protein levels on the serum biochemistry of donkeys. In the MP group, the concentrations of ALB and HDL were significantly higher (*p* < 0.05) compared to the LP and HP groups. Additionally, as the dietary crude protein level increased, the urea content was markedly elevated (*p* < 0.05) in both the MP and HP groups relative to the LP group. However, no significant differences (*p* > 0.05) were noted in other serum biochemistry parameters across the three groups.

### 3.3. The Dietary Nutrient Digestibility of the Donkeys

Table 4 presents the digestibility of dietary nutrients in donkeys across the three groups. As the dietary protein level increased, the digestibility of CP, NDF, HCEL, OM, and NEF initially increased and then decreased, with the highest values (*p* < 0.05) observed in the MP group. No significant differences were observed in the digestibility of DM, Ash, EE, CF, ADF, CEL, and CH_2_O (*p* > 0.05).

### 3.4. The Slaughter Performance and Meat Quality Traits of the Donkeys

Table 5 displays the slaughter performance and meat quality of the donkeys in the three groups. Carcass weight, dressing percentage, shearing force, and organic matter (OM) content in the meat of the MP group were improved (*p* < 0.05) compared to those in the LP and HP groups. DM and EE levels in the HP group were lower (*p* < 0.05) than those in the MP and LP groups. Additionally, in this study, dietary protein levels had no influence (*p* > 0.05) on other indices of slaughter performance and meat quality traits of the donkeys.

### 3.5. Amino Acids in the Meat of the Donkeys

Table 6 exhibits the impact of varying dietary protein levels on the amino acid composition in donkey meat. Notably, the MP group exhibited significantly higher concentrations of Ile, NEAAs, and FAAs (*p* < 0.05) compared to both the LP and HP groups. The Asp level in the LP group was significantly lower (*p* < 0.05) than that in the MP group, whereas no significant difference (*p* > 0.05) was observed between the LP and HP groups.

### 3.6. Fatty Acids in the Meat of the Donkeys

Table 7 presents the fatty acid composition of meat from donkeys across the three groups. In this study, palmitic (C16:0), stearic (C18:0), and myristic acids (C14:0) were identified as the primary saturated fatty acids (SFAs) in the longissimus dorsi muscle. The levels of C16:0 and C14:0 were significantly higher (*p* < 0.05) in the LP and MP groups compared to the HP group. Similarly, the levels of hexanoic acid (C6:0), pentadecanoic acid (C15:0), and heptadecanoic acid (C17:0) in meat progressively decreased as the dietary protein levels increased. Notably, the level of C15:0 in the MP group was significantly higher (*p* < 0.05) than in the other two groups. In contrast, significant differences (*p* < 0.05) in the levels of C6:0 and C17:0 were only observed between the LP and HP groups. However, no significant differences (*p* > 0.05) were detected among the three groups regarding the level of C18:0. The predominant monounsaturated fatty acids (MUFAs) across all donkeys in both groups were oleic (C18:1n9c), elaidic (C18:1n9t), and palmitoleic acids (C16:1). Compared to the MP and LP groups, the levels of these MUFAs were significantly lower (*p* < 0.05) in the HP group. Linoleic acid (C18:2n6c) was the principal polyunsaturated fatty acid (PUFA), with no significant differences (*p* > 0.05) observed among the three groups. Furthermore, the levels of total fatty acids (TFAs), PUFAs, MUFAs, unsaturated fatty acids (UFAs), and SFAs gradually decreased as the protein levels increased, with the decrease being particularly significant (*p* < 0.05) in the HP group.

### 3.7. Gut Microbiota

#### 3.7.1. Diversity Analysis of the Rectal Microbes

Table 8 lists the data on good coverage, indicating that the sequencing depth coverage rate surpassed 99% for all microbial species.

As illustrated in Figure 1A, based on a 97% sequence similarity threshold, the LP group contained a total of 2517 operational taxonomic units (OTUs), which were classified into 25 phyla, 46 classes, 89 orders, 143 families, and 253 genera. In the MP group, 2442 OTUs were identified and clustered into 25 phyla, 47 classes, 89 orders, 147 families, and 233 genera. The HP group had 2553 OTUs distributed across 26 phyla, 47 classes, 84 orders, 136 families, and 227 genera. A total of 2049 OTUs were shared among all three groups.

Table 8 also exhibits the alpha-diversity indices for the three groups. The OS, Chao1, and ACE indices progressively increased with rising dietary crude protein levels, and all were significantly higher (*p* < 0.05) in the HP group compared to the LP group. Additionally, the Shannon index indicates that the HP group was significantly higher (*p* < 0.05) than the MP group. No other significant differences (*p* > 0.05) were observed among the alpha-diversity indices.

Figure 1B illustrates the beta-diversity of rectal microbial communities in this study through PCoA analysis, demonstrating that these communities were primarily separated based on varying levels of dietary crude protein.

#### 3.7.2. Profiles of the Rectal Microbes

Figure 1C–F present the profile parameters of rectal microbes (relative abundance > 1.0%) in this study. At the phylum level, *Firmicutes* and *Bacteroidetes* were the predominant colonizing microbes across all groups. Specifically, the relative abundances of *Firmicutes* were 53.71%, 49.94%, and 53.56%, while those of *Bacteroidetes* were 32.08%, 39.20%, and 31.41% in the three groups, respectively. The *Firmicutes*-to-*Bacteroidetes* (F/B) ratios were calculated as 1.79, 1.31, and 1.78 for each group. Within the phylum *Firmicutes*, the dominant families included *Streptococcaceae* and *Oscillospiraceae*, whereas, within the phylum *Bacteroidota*, the main families were *Rikenellaceae*, *Bacteroidaceae*, and *F082*. At the genus level, the top 10 microbial communities across all groups were *Rikenellaceae*_*RC9*_*gut*_*group*, *Streptococcus*, *Bacteroides*, *Treponema*, *NK4A214*_*group*, *Allobaculum*, *Oscillospiraceae*_*UCG*-*002*, *Fibrobacter*, *Prevotellaceae*_*UCG*-*001*, and *Prevotellaceae*_*UCG*-*004*.

#### 3.7.3. Discrepant Microbe Analysis

Figure 2A–E illustrate the changes in microbial communities (relative abundance > 1.0% and TOP 20) at the genus level due to different dietary protein levels, as analyzed by one-way *ANOVA*. In the MP group, the relative abundances of the *NK4A214*_*group*, *Oscillospiraceae*_*UCG*-*002*, and *Oscillospiraceae*_*UCG*-*005* were significantly lower (*p* < 0.05) compared to those in the HP group, while no significant differences (*p* > 0.05) were observed relative to the LP group. Conversely, *Clostridium*_*sensu*_*stricto*_*1* showed a significant increase (*p* < 0.05) in the MP group compared to both the LP and HP groups. Additionally, *Prevotella* levels in the LP group were significantly higher (*p* < 0.05) than those in both the MP and HP groups.

Additionally, the linear discriminant analysis effect size (LEfSe) analysis, with an LDA score threshold > 3.5, identified significant microbial differences among the three groups. At the genus level, *Prevotella* and *Alloprevotella* were the predominant microbial species in the LP group. The MP group was characterized by a high abundance of *Clostridium*_*sensu*_*stricto*_*1*, while the HP group exhibited the notable presence of the *NK4A214*_*group*, *Oscillospiraceae*_*UCG*-*002*, and *Oscillospiraceae*_*UCG*-*005*. The cladograms illustrate the phylogenetic distribution of these differentially abundant bacteria across the groups.

By integrating the two above-mentioned discrepant microbe analysis methods, we conclude that the biomarkers of microbial changes attributable to different dietary protein levels are the *NK4A214*_*group*, *Oscillospiraceae*_*UCG*-*002*, *Clostridium*_*sensu*_*stricto*_*1*, *Prevotella*, and *Oscillospiraceae*_*UCG*-*005*. Additionally, Figure 2H illustrates the correlations among the relative abundances of the five distinct microorganisms, revealing a negative correlation between *Clostridium*_*sensu*_*stricto*_*1* and *Prevotella* with the *NK4A214*_*group*, *Oscillospiraceae*_*UCG*-*005*, and *Oscillospiraceae*_*UCG*-*002*.

## 4. Discussion

Dietary protein levels significantly influence the growth, fattening performance, and meat quality of animals, thereby enhancing their overall growth performance. Current research on protein nutrition predominantly centers on livestock such as cattle, sheep, and goats. Studies have demonstrated that optimal dietary protein levels improve average daily gain and feed efficiency in steers, heifers, and lambs [28,29]. Research on the dietary protein requirements of donkeys remains notably limited. The findings of this study demonstrate that dietary protein levels of 12.52% (MP) and 14.06% (HP) improved growth performance, as evidenced by improved ADG and FCRs compared to donkeys fed 11.03% (LP) dietary protein. This finding aligns with previous research, which has demonstrated that an inadequate dietary protein supply can significantly limit the production performance of lambs, particularly their ADG [30]. Additionally, the results of the current study are consistent with earlier reports indicating that reduced dietary protein may improve the FCR [31]. Inadequate dietary protein impairs the ability of livestock to maintain organ function and hinders tissue formation or renewal [32], such as muscle or skin in Dezhou donkeys, during the fattening period. Additionally, carbohydrates and fats cannot be effectively converted into amino acids through endogenous metabolic processes. Therefore, to optimize production performance, it is crucial to ensure adequate levels of dietary protein for these animals during this critical growth phase.

The dietary protein level not only satisfies the requirements for animal fattening but also contributes to their overall health. In our study, the level of serum ALB and HDL were both improved for donkeys fed a diet with 12.52% (MP) compared with the donkeys fed 14.06% (HP) and 11.03% (LP) dietary protein. However, the serum urea in 14.06% (HP) and 12.52% (MP) was higher than that in 11.03% (LP). The ALB plays a crucial role in exerting potent antioxidant and anti-inflammatory effects [33]. Elevated dietary protein intake has been shown to positively correlate with increased serum triglycerides [34] and high-density lipoprotein levels (HDLs) [35], indicating that adequate protein intake can effectively regulate lipid metabolism and promote fat deposition. Serum urea levels serve as an indicator of the ability of livestock to utilize dietary nitrogen [36], which may result in elevated nitrogen emissions due to a high-protein diet [37]. Meanwhile, this parameter also signifies an enhancement in the nutritional status of donkeys [38]. However, an excessive supply of dietary protein can lead to an overabundance of amino acids, thereby increasing the metabolic burden on the liver.

The equine hindgut, comprising the cecum and colon, constitutes approximately 60% of the gastrointestinal tract. This region harbors a vast microbial community that plays an essential role in the digestion of fibrous feeds and supports overall nutritional metabolism [39]. The composition of hindgut microbes in donkeys can be modulated by adjusting dietary protein levels [40]. In this study, α-diversity analysis demonstrates a concentration-dependent increase in the rectal bacterial community across different dietary protein levels. The elevated nitrogen content likely promotes nutrient enrichment, thereby enhancing α-diversity within the donkey intestine. The β-diversity analysis reveals distinct clustering patterns with partial overlap in the rectal microbiota among the three groups, indicating both differences and similarities in microbial composition. In particular, the general *NK4A214*_*group*, *Oscillospiraceae*_*UCG*-*005*, and *Oscillospiraceae*_*UCG*-*002* exhibited low relative abundance in the MP group, whereas *Clostridium*_*sensu*_*stricto*_1 was obviously enhanced. The *NK4A214*_*group*, *Oscillospiraceae*_*UCG*-*002*, and *Oscillospiraceae*_*UCG*-*005* have been reported to be enriched in animals fed high-fiber diets [41] and increased from pre-weaning to post-weaning in the hindgut of calves and donkeys [42,43]. As the predominant genus in the donkey hindgut, *Clostridium*_*sensu*_*stricto*_*1* is crucial for maintaining intestinal balance and enhancing dietary dry matter digestibility and antioxidant capacity in donkeys [44]. This bacterium efficiently breaks down indigestible nutrients, producing significant amounts of volatile fatty acids (VFAs) [45,46]. Furthermore, *Prevotella*, belonging to the *Prevotellaceae* family, commonly assists in the breakdown of dietary protein and carbohydrates, especially the capacity of the gut microbiota to ferment complex indigestible carbohydrates from diets [47]. *Prevotella*, in our study, displayed growth advantages in the protein-rich environment of the donkey gastrointestinal tract, and the relative abundance clearly decreased in the HP group. Additionally, the relative abundance of *Clostridium*_*sensu*_*stricto*_*1* and *Prevotella* displays a negative correlation with the *NK4A214*_*group*, *Oscillospiraceae*_*UCG*-*005*, and *Oscillospiraceae*_*UCG*-*002*. Consistent with these trends in the rectal microbiota at the genus level, the digestibility of CF, NDF, HCEL, OM, and NFE was also observed in the MP group. This potentially enhances dietary protein digestibility, which suggests a positive impact on amino acid metabolism. However, limited research has reported the association of these microbes with amino acid metabolism, and further investigation is required to elucidate the underlying mechanism.

The meat quality of donkeys, including the cooking rate, water-holding capacity, color, and nutrient content, such as protein and fat, is influenced by various factors like animal breed, farm management, and nutrition [39]. Previous studies have indicated that the increasing levels of maternal dietary protein for cows enhanced both the dressing percentage and meat tenderness of their offspring calves [48]. Similarly, a dietary protein level of 12.99% has been observed to positively impact the tenderness of mutton [11]. Our results also indicate that a dietary protein content of 12.52% (MP) enhances the dressing percentage, tenderness, and nutrient level, including the DM and OM of the longissimus dorsi muscle. Meat tenderness, as assessed by shear force analysis, is positively correlated with intramuscular fat content [49,50]. In our study, the MP group exhibited a higher crude fat content compared to both the LP and HP groups, which supports this correlation. Donkey meat is characterized by high contents of crude protein but low total fat, cholesterol, and calories, which is greatly appreciated by consumers [2]. Nevertheless, from an alternative perspective, the lower fat content partially contributes to a relatively tougher texture upon consumption. Given that consumer preference often favors meat with optimal tenderness, it is necessary to appropriately enhance the tenderness of donkey meat through optimized animal diets.

Donkey meat appears to be an important source of high-biological-value proteins as it contains essential amino acids in good amounts [51]. Various dietary treatments also exerted an influence on the flavor of donkey meat by facilitating the generation of amino acids in the muscle, thereby enhancing its sensory characteristics and improving taste [52]. Specifically, flavor-contributing amino acids such as arginine, lysine, aspartic acid, and glutamic acid undergo Maillard reactions by reacting with soluble reducing sugars during the heating process, thereby serving as precursors to flavor compounds [53]. Additionally, leucine also serves as a flavor precursor [54] when subjected to the Maillard reaction [55]. Currently, research on the relationship between dietary protein and meat amino acid composition remains limited. In this study, the contents of FAAs, NEAAs, Ile, and Asp were elevated in the MP group, suggesting that the donkey meat flavor was influenced by the level of dietary protein. Additionally, among the three groups, lysine was identified as the most abundant EAA followed by leucine. In contrast, Asp and Glu were found to be the predominant amino acids in the NAA group. Future studies should further investigate the underlying mechanisms. Moreover, the medium protein level can, therefore, be concluded to enhance the umami content of donkey meat, while the levels of lysine and methionine decreased; future studies should further investigate the underlying mechanisms. This is according to the results described in [51].

In our study, the fatty acid profile of the longissimus dorsi muscle varied among different groups. For instance, the content of C16:0 exhibited a decreasing trend with increasing dietary protein levels, although no significant differences were observed between the LP and MP groups. Compared to the MP and LP groups, the HP group showed reduced levels of C18:1n9c, C18:1n9t, C16:1, TFAs, PUFAs, MUFAs, and SFAs. Similar to our study, a level of 16.8% of CP in the diet increased the percentage of MUFAs in the intramuscular fat of steers [56]. Furthermore, suckling techniques, including artificial and natural, also influenced the intramuscular fat acid composition of the longissimus dorsi muscle, and artificially reared donkeys showed a more favorable profile in meat [57]. Notably, during cooking processes, fatty acids, like C16:0, C18:0, and C18:1, undergo oxidative degradation [58,59,60], leading to the production of significant amounts of unsaturated aldehydes that contribute to a diverse range of flavors [61,62,63]. Moreover, the contents of C16:1 were positively correlated to better tenderness, juiciness, and flavor of meat [64,65]. The higher protein diet is likely attributed to the downregulation of the fatty acid synthase gene, resulting in the diminished synthesis of medium- and long-chain fatty acids [66]. Interestingly, donkey meat exhibits a favorable fatty acid profile, particularly with respect to the relatively ideal proportions of unsaturated fatty acids [67].

## 5. Conclusions

In this study, we investigated the effects of varying dietary protein levels on growth and slaughter performance, meat quality traits, amino acid and fatty acid profiles, and hindgut microbial composition in Dezhou donkeys. The results indicate that a 12.52% protein diet significantly enhanced nutrient digestibility, including protein and NDF, leading to improvements in ADG, FCRs, carcass weight, and dressing percentage. Higher dietary protein levels also positively influenced hindgut microbiota diversity, particularly by increasing the abundance of *Clostridium*_*sensu*_*stricto*_1 and *Prevotella* at 12.52% dietary protein, which is crucial for fiber degradation, energy metabolism, and nitrogen utilization. Additionally, changes in tenderness, amino acid, and fatty acid compositions contributed to improved meat quality and flavor. Our findings provide valuable insights into the microbiota dynamics of fattening Dezhou donkeys at different protein intake levels and characterize the impact of protein intake on meat quality and flavor, offering new perspectives for optimizing dietary protein levels for fattening donkeys.

## Figures and Tables

**Figure 1 microorganisms-13-01388-f001:**
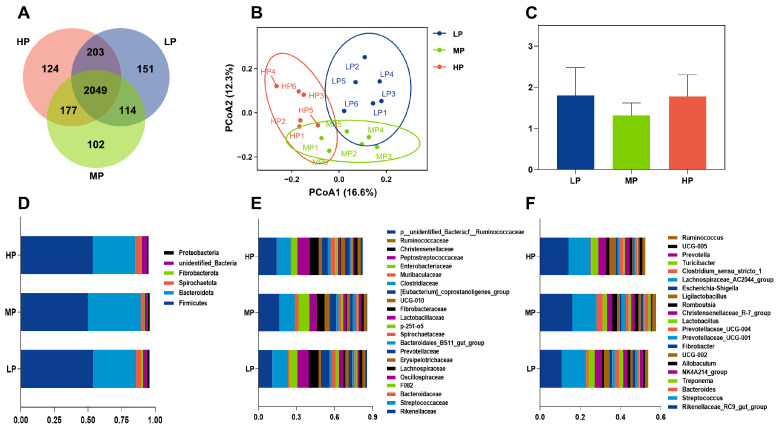
Effects of varying dietary protein levels on the composition and relative abundance of the rectal microbial community in donkeys in the three groups. (**A**) The shared and distinct OTUs among the three groups; (**B**) Principal Coordinate Analysis (PCoA) plot of beta-diversity; (**C**) the ratio of *Firmicutes* to *Bacteroidetes*; the relative abundances at the phylum (**D**), family (**E**), and genus (**F**) levels.

**Figure 2 microorganisms-13-01388-f002:**
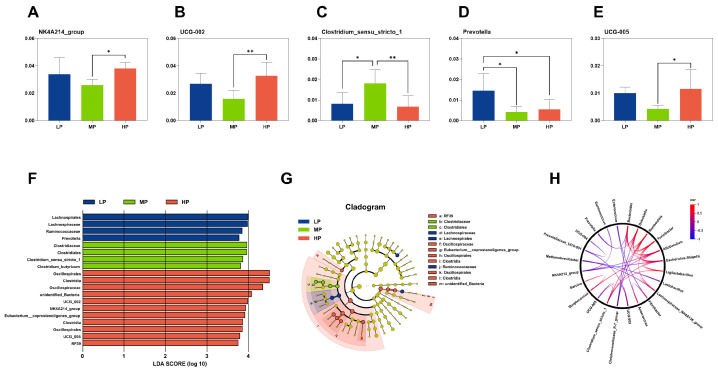
One-way ANOVA analysis was conducted on microbial genera with relative abundances > 1.0% (**A**–**E**). Through the LefSe analysis (LDA score > 3.5), discrepant rectum microorganisms among the three groups are visualized in a histogram (**F**) and a cladogram (**G**), illustrating their increase or decrease status. Additionally, from the innermost to the outermost layer of the cladogram, the species classifications are represented at the phylum, class, order, family, and genus levels. The relationships among the top 30 microbes at the genus level (**H**) and correlation coefficients greater than 0.5 are displayed. Blue lines indicate negative correlations, while red lines represent positive correlations. Statistical significance was defined as * *p* < 0.05 or ** *p* < 0.01.

**Table 1 microorganisms-13-01388-t001:** Basic diet composition and nutritional level (air-dried basis).

Items	Treatments ^1^		
	LP	MP	HP
Feed composition			
Corn, %	26.00	22.00	17.50
Soybean meal, %	1.00	5.00	9.50
Wheat middling, %	35.00	35.00	35.00
Soybean oil, %	0.60	0.60	0.60
Peanut vine, %	35.00	35.00	35.00
Limestone, %	0.40	0.40	0.40
CaHPO_4_, %	0.55	0.55	0.55
Lysine (Lys), %	0.20	0.20	0.20
Methionine, %	0.10	0.10	0.10
NaHCO_3_, %	0.30	0.30	0.30
Premix ^2^, %	0.45	0.45	0.45
NaCl, %	0.40	0.40	0.40
Total, %	100.00	100.00	100.00
Nutrient levels ^3^			
Dry matter (DM), %	88.36	88.50	88.91
Digestible energy (DE), MJ/kg	11.81	11.79	11.77
Crude protein (CP), %	11.03	12.52	14.06
Ether extract (EE), %	3.01	2.95	2.87
Lysine (Lys), %	0.49	0.59	0.70
Crude fiber (CF), %	14.99	16.12	16.20
Neutral detergent fiber (NDF), %	34.23	34.35	34.49
Acid detergent fiber (ADF), %	17.69	18.77	19.14
Acid detergent lignin (ADL), %	4.80	4.55	4.52
Hemicellulose (HCEL), %	16.54	15.58	15.35
Ash, %	7.56	8.30	7.65
Organic matter (OM)	80.80	80.20	81.48
Carbohydrate (CH_2_O)	66.76	64.73	64.55
Nitrogen-free extract (NFE)	51.77	48.61	48.35
Calcium (Ca), %	0.62	0.63	0.64
Phosphorus (P), %	0.54	0.55	0.57

^1^ Donkeys were fed diets containing 11.03% (LP), 12.52% (MP), and 14.06% (HP) crude protein, respectively. ^2^ Premix: Each kilogram of the premix contains VA 8000 IU, VD 2400 IU, VE 30 mg, VK3 3 mg, VB1 3 mg, VB2 8 mg, VB3 10.8 mg, VB5 34 mg, VB6 4 mg, VB9 1.4 mg, VB12 0.02 mg, VB7 0.13 mg, Cu 8.3 mg, Fe 60 mg, Zn 70 mg, Mn 62 mg, Se 0.25 mg, and I 0.5 mg. ^3^ Nutrient levels: DE in the formula table is the calculated value, and other nutritional components are the measured values.

**Table 2 microorganisms-13-01388-t002:** Effects of diets with different protein levels on donkey production performance.

Items	Treatments ^1^			SEM	*p*-Value
	LP	MP	HP		
Initial BW ^2^, kg	188.33	188.25	188.42	2.415	0.998
Final BW ^3^, kg	220.42	223.08	224.07	2.631	0.381
ADG ^4^, g/d	534.72 ^b^	580.56 ^ab^	594.17 ^a^	22.490	0.045
DMI ^5^, kg/d	5.28	5.30	5.28	0.092	0.978
FCR ^6^	9.90 ^a^	9.18 ^b^	8.90 ^b^	0.249	0.003

Means marked with different superscripts within the same row are significantly different (*p* < 0.05). ^1^ Donkeys were fed diets containing 11.03% (LP), 12.52% (MP), and 14.06% (HP) crude protein, respectively. ^2^ Initial BW, initial body weight. ^3^ Final BW, final body weight. ^4^ ADG, average daily gain. ^5^ DMI, average daily dry matter intake. ^6^ FCR, feed conversion ratio, is calculated by dividing DMI by ADG.

**Table 3 microorganisms-13-01388-t003:** Effects of diets with different protein levels on donkey serum biochemistry parameters.

Items	Treatments ^1^			SEM	*p*-Value
	LP	MP	HP		
Glutamic pyruvic transaminase (ALT), U/L	6.67	8.00	8.00	1.393	0.556
Glutamic oxaloacetic transaminase (AST), U/L	215.17	226.76	210.83	17.627	0.654
AST/ALT	34.83	31.59	28.84	6.573	0.666
Alkaline phosphatase (ALP), U/L	168.00	150.50	172.67	22.178	0.585
Total protein (TP), mmol/L	59.79	58.92	58.94	0.133	0.688
Albumin (ALB), mmol/L	13.57 ^b^	14.67 ^a^	14.10 ^b^	0.415	0.034
Urea, mmol/L	7.74 ^b^	9.18 ^a^	10.02 ^a^	0.637	0.009
Uric acid (UA), μmol/L	2.67	3.33	2.67	0.596	0.454
Creatine kinase (CK), U/L	142.44	135.17	141.33	22.218	0.940
Glucose (GLU), mmol/L	4.99	492	5.17	0.366	0.782
Triglyceride (TG), mmol/L	0.44	0.49	0.52	0.072	0.524
Total cholesterol (TCHO), mmol/L	1.39	1.53	1.43	0.092	0.305
High density lipoprotein (HDL), mmol/L	0.98 ^b^	1.16 ^a^	0.99 ^b^	0.091	0.023
Low density lipoprotein (LDL), mmol/L	0.17	0.18	0.15	0.028	0.575
Lactic dehydrogenase (LDH), U/L	306.50	342.83	286.17	45.651	0.471
Calcium (CA), mmol/L	4.30	4.31	4.33	0.085	0.945
Phosphorus (P), mmol/L	1.60	1.68	1.57	0.083	0.397

Means marked with different superscripts within the same row are significantly different (*p* < 0.05). ^1^ Donkeys were fed diets containing 11.03% (LP), 12.52% (MP), and 14.06% (HP) crude protein, respectively.

**Table 4 microorganisms-13-01388-t004:** Effects of diets with different protein levels on the digestibility of donkeys.

Items	Treatments ^1^			SEM	*p*-Value
	LP	MP	HP		
Dry matter (DM), %	61.91	62.79	58.49	2.370	0.194
Ash, %	28.88	27.69	30.28	3.722	0.787
Ether extract (EE), %	72.54	75.87	71.05	4.315	0.534
Crude protein (CP), %	69.14 ^c^	78.95 ^a^	75.12 ^b^	1.488	<0.01
Crude fiber (CF), %	26.79	26.34	24.71	2.621	0.710
Neutral detergent fiber (NDF), %	37.54 ^b^	40.24 ^a^	38.85 ^ab^	0.932	0.036
Acid detergent fiber (ADF), %	31.34	30.27	30.29	1.272	0.640
Hemicellulose (HCEL), %	48.35 ^b^	58.47 ^a^	52.70 ^ab^	3.242	0.023
Organic matter (OM), %	71.57 ^ab^	72.41 ^a^	69.50 ^b^	1.095	0.048
Carbohydrate (CH_2_O), %	71.95	70.80	70.69	1.454	0.640
Nitrogen-free extract (NFE), %	82.92 ^ab^	84.00 ^a^	81.62 ^b^	0.748	0.021

Means marked with different superscripts within the same row are significantly different (*p* < 0.05). ^1^ Donkeys were fed diets containing 11.03% (LP), 12.52% (MP), and 14.06% (HP) crude protein, respectively.

**Table 5 microorganisms-13-01388-t005:** Effects of dietary protein levels on the slaughter performance and meat quality traits of donkeys.

Items	Treatments ^1^			SEM	*p*-Value
	LP	MP	HP		
Slaughter Performance					
Live weight, kg	226.00	226.64	226.40	1.606	0.990
Carcass weight, kg	123.15 ^b^	132.81 ^a^	122.72 ^b^	1.622	0.004
Dressing percentage, %	54.49 ^b^	58.60 ^a^	54.23 ^b^	0.760	0.017
Meat color					
Lightness, L*	30.43	31.23	31.48	0.635	0.809
Redness, a*	17.11	17.63	18.01	0.544	0.825
Yellowness, b*	8.53	8.90	8.93	0.324	0.878
Meat quality					
Drip loss, %	1.48	1.41	1.46	0.031	0.853
Cooking loss, %	40.91	40.60	39.28	0.756	0.689
Shearing force, N	80.75 ^a^	73.36 ^b^	80.88 ^a^	2.674	0.032
Meat nutrients (fresh sample basis)					
Dry matter (DM), %	28.38 ^a^	28.58 ^a^	28.08 ^b^	0.120	0.008
Crude protein (CP), %	23.45	23.01	23.63	0.591	0.576
Crude fat (EE), %	3.05 ^ab^	3.81 ^a^	2.48 ^b^	0.360	0.015
Ash, %	1.58	1.37	1.61	0.257	0.626
Organic matter (OM), %	26.57 ^b^	27.20 ^a^	26.48 ^b^	0.229	0.025

Means marked with different superscripts within the same row are significantly different (*p* < 0.05). ^1^ Donkeys were fed diets containing 11.03% (LP), 12.52% (MP), and 14.06% (HP) crude protein, respectively.

**Table 6 microorganisms-13-01388-t006:** Effects of diets with different protein levels on the amino acids of donkey meat (mg/100 g).

Items	Treatments ^1^			SEM	*p*-value
	LP	MP	HP		
EAAs					
Histidine (His)	1266.01	1361.20	1217.55	10.586	0.417
Threonine (Thr)	1285.17	1381.64	1328.78	115.001	0.716
Lysine (Lys)	4776.79	4415.28	4477.30	211.685	0.206
Methionine (Met)	437.92	369.74	450.86	74.348	0.539
Valine (Val)	1417.16	1515.71	1436.92	139.307	0.765
Isoleucine (Ile)	1463.22	1568.02	1465.89	76.101	0.356
Leucine (Leu)	2754.70 ^b^	2958.52 ^a^	2798.08 ^b^	57.617	0.027
Phenylalanine (Phe)	1313.73	1378.58	1306.66	55.086	0.411
NEAAs					
Alanine (Ala)	1979.19	2074.27	2028.73	126.824	0.764
Proline (Pro)	1664.18	1746.19	1708.51	184.941	0.908
Cystine (Cys)	62.00	60.64	63.53	10.272	0.961
Tyrosine (Tyr)	879.17	938.76	917.01	35.827	0.313
Serine (Ser)	1061.09	1154.84	1136.24	100.144	0.634
Glycine (Gly)	1426.67	1477.03	1447.72	134.730	0.933
Aspartic acid (Asp)	2734.84 ^b^	2924.00 ^a^	2823.43 ^ab^	58.657	0.049
Glutamic acid (Glu)	2461.85	2651.50	2601.12	94.563	0.197
TAAs ^2^	26,983.68	27,918.70	27,208.32	611.373	0.348
EAAs ^3^	14,714.68	14,891.46	14,482.03	325.894	0.510
NEAAs ^4^	12,269.00 ^b^	13,027.24 ^a^	12,726.29 ^b^	199.808	0.025
FAAs ^5^	8602.55 ^b^	9126.80 ^a^	8901.00 ^b^	151.625	0.037

Means marked with different superscripts within the same row are significantly different (*p* < 0.05). ^1^ Donkeys were fed diets containing 11.03% (LP), 12.52% (MP), and 14.06% (HP) crude protein, respectively. ^2^ TAAs: total amino acids. ^3^ EAAs: essential amino acids. ^4^ NEAAs: non-essential amino acids (sum of Asn, Asp, and Glu). ^5^ FAAs: flavor amino acids (sum of Glu, Asp, Ala, and Gly).

**Table 7 microorganisms-13-01388-t007:** Effects of diets with different protein levels on the fatty acids of donkey meat (mg/100 g).

Items	Treatments ^1^			SEM	*p*-Value
	LP	MP	HP		
C6:0	0.08 ^a^	0.06 ^ab^	0.04 ^b^	0.010	0.027
C8:0	0.64	0.85	0.39	0.200	0.153
C10:0	0.42	0.42	0.28	0.117	0.436
C11:0	0.05	0.05	0.03	0.009	0.149
C12:0	1.26	1.16	0.81	0.306	0.359
C13:0	0.04	0.03	0.02	0.008	0.125
C14:0	19.42 ^a^	15.79 ^a^	10.42 ^b^	2.116	0.015
C15:0	1.14 ^a^	0.81 ^b^	0.58 ^b^	0.126	0.012
C16:0	253.77 ^a^	241.61 ^a^	181.81 ^b^	21.167	0.030
C17:0	1.86 ^a^	1.52 ^ab^	1.12 ^b^	0.207	0.031
C18:0	93.51	92.55	81.76	8.558	0.374
C20:0	1.06	1.08	0.99	0.148	0.802
C21:0	0.26	0.20	0.17	0.046	0.234
C22:0	0.44	0.53	0.45	0.058	0.302
C23:0	0.21	0.21	0.21	0.012	0.865
C24:0	0.14	0.18	0.15	0.028	0.459
C14:1	1.19	1.00	0.62	0.220	0.100
C16:1	27.09 ^a^	26.29 ^a^	15.50 ^b^	3.535	0.029
C18:1n9t	112.84 ^a^	103.59 ^a^	55.85 ^b^	17.864	0.039
C18:1n9c	122.85 ^a^	117.40 ^a^	71.49 ^b^	18.171	0.043
C20:1	2.02 ^a^	1.76 ^a^	1.16 ^b^	0.235	0.027
C22:1n9	1.34 ^b^	2.28 ^a^	1.82 ^ab^	0.285	0.045
C18:2n6c	39.25	35.46	31.59	7.357	0.608
C22:2	0.26 ^a^	0.20 ^b^	0.20 ^b^	0.020	0.041
C20:3n3	0.09	0.10	0.07	0.015	0.281
C20:4n6	1.64	1.20	2.08	0.401	0.538
C22:6n3	0.05	0.06	0.05	0.005	0.249
TFAs ^2^	682.91 ^a^	647.17 ^a^	459.64 ^b^	72.175	0.044
PUFAs ^3^	41.28	37.80	33.99	7.638	0.654
MUFAs ^4^	267.34 ^a^	252.32 ^a^	146.43 ^b^	33.629	0.022
UFAs ^5^	308.62 ^a^	290.12 ^a^	180.43 ^b^	40.790	0.040
SFAs ^6^	374.28 ^a^	357.04 ^a^	279.21 ^b^	26.519	0.025
PUFAs/SFAs	0.11	0.11	0.12	0.013	0.524
n-3 ^7^	0.14	0.15	0.12	0.016	0.189
n-6 ^8^	40.89	37.45	33.67	7.607	0.657

Means marked with different superscripts within the same row are significantly different (*p* < 0.05). ^1^ Donkeys were fed diets containing 11.03% (LP), 12.52% (MP), and 14.06% (HP) crude protein, respectively. ^2^ TFAs: total fatty acids (sum of all the determined fatty acids); ^3^ PUFAs: polyunsaturated fatty acids; ^4^ MUFAs: monounsaturated fatty acids; ^5^ UFAs: unsaturated fatty acids; ^6^ SFAs: saturated fatty acids (sum of PUFAs and MUFAs); ^7^ n-3: Omega-3 fatty acids (sum of C20:3n3 and C22:6n3); ^8^ n-6: Omega-6 fatty acids (sum of C18:2n6 and C20:4n6).

**Table 8 microorganisms-13-01388-t008:** Alpha-diversity of the gut microbiota of donkey fed diets with diff. CP levels.

Items	Treatments ^1^			SEM	*p*-Value
	LP	MP	HP		
Good coverage, %	99.62	99.55	99.55	0.001	0.404
Observed species (OS)	1427.00 ^b^	1525.00 ^ab^	1623.00 ^a^	69.174	0.040
Chao1	1524.34 ^b^	1663.19 ^ab^	1743.90 ^a^	66.980	0.016
ACE	1549.89 ^b^	1686.77 ^ab^	1771.04 ^a^	75.256	0.031
PD_whole_tree	106.92	111.97	116.69	6.535	0.388
Shannon	7.65 ^ab^	7.46 ^b^	7.97 ^a^	0.185	0.040
Simpson	0.97	0.97	0.98	0.008	0.509

Means marked with different superscripts within the same row are significantly different (*p* < 0.05). ^1^ Donkeys were fed diets containing 11.03% (LP), 12.52% (MP), and 14.06% (HP) crude protein, respectively.

## Data Availability

The microbiome datasets presented in this study can be found in the online repositories (the accession number can be found below) (PRJNA977424) at https://www.ncbi.nlm.nih.gov/sra/ (accessed on 30 May 2023). The other data presented in this study are available on request from the corresponding authors due to privacy or ethical restrictions.

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
