# Peer review of "Effects of Dietary Protein Levels on Production Performance, Meat Quality Traits, and Gut Microbiome of Fatting Dezhou Donkeys"

_microorganisms, 2025, doi:10.3390/microorganisms13061388_

Round 1
Reviewer 1 Report
Comments and Suggestions for Authors
Dear authors
I am pleased to be invited to read this interesting manuscript. please check comments on the attached documents.

Reviewer 2 Report
Comments and Suggestions for Authors
This study aimed to investigate the effects of varying dietary protein levels on growth performance, meat quality traits, amino acid and fatty acid compositions, and hindgut microbiota in Dezhou donkeys. The introduction chapter contains a review of the world literature on the subject of the article. The amount of donkeys used in these studies is relatively small, there is no information about replicates for growth performance, for the remaining characteristics these were individual animals. The research methods used are correct. The discussion is exhaustively correct.
General comments
The research methods used are correct. The Abstract, Materials and Methods, and Results chapters requires additions in some places
No information on the type of building (closed, no runs, no windows?), housing system and type of litter
No information on temperature, relative humidity, the lighting program (length, intensity, color, type)
What scale was used to determine BW, FI
No information on the study of the normal distribution of traits
Other
Please use a small p In italic (p < 0.05) instead of a capital P
When giving the temperature, a space should be used after the number, for example -20 °C instead of -20°C
The Reference chapter must be prepared in accordance with the requirements of Microorganisms journal. please use a "dot" after each abbreviation, position No. 3, 7, 9, 12, 16, 17, 30, 32, 33, 34,46, 53, 59, 63, and 66
Detailed comments
L9 please add a phone number to the corresponding author
L13 what age are the donkeys?
L16 MP also - see Table 2
L18-19 no significant differences with the LP group
L20+ no description of the effect of diet on basic chemical composition and serum parameters
Table 1 from page 3
L94, 95 „method” with a lower case letter
L99-102 Is there data for ash percentage and other listed components in the diet in this article?
L151 Please provide calibration plate data
L284 "Shearing" in lower case
L286 no OM description
Table 5 from page 9
In Table 5 L* In italic
a*, lower case In italic instead of current form
b* , lower case In italic instead of current form
L315 no description for C6:0; C15:0; C17:0
L349 Or LP as well?
L526 [64,65] instead of current form
